# MutaPLM: Protein Language Modeling for Mutation Explanation and Engineering

**Yizhen Luo**[1,2,†], **Zikun Nie**[1,2,†], **Massimo Hong**[1,2],
**Suyuan Zhao**[1,2], **Hao Zhou**[1,*], **Zaiqing Nie**[1,3,*]
[1]Institute of AI Industry Research (AIR), Tsinghua University
[2]Department of Computer Science and Technology, Tsinghua University
[3]Pharmolix Inc.
`{yz-luo22,nzk24,hongcd21,zhaosy23}@mails.tsinghua.edu.cn`
`{zhouhao,zaiqing}@air.tsinghua.edu.cn`

## Abstract

Studying protein mutations within amino acid sequences holds tremendous significance in life sciences. Protein language models (PLMs) have demonstrated strong capabilities in broad biological applications. However, due to architectural design and lack of supervision, PLMs model mutations implicitly with evolutionary plausibility, which is not satisfactory to serve as explainable and engineerable tools in real-world studies. To address these issues, we present MutaPLM, a unified framework for interpreting and navigating protein mutations with protein language models. MutaPLM introduces a protein *delta* network that captures explicit protein mutation representations within a unified feature space, and a transfer learning pipeline with a chain-of-thought (CoT) strategy to harvest protein mutation knowledge from biomedical texts. We also construct MutaDescribe, the first large-scale protein mutation dataset with rich textual annotations, which provides cross-modal supervision signals. Through comprehensive experiments, we demonstrate that MutaPLM excels at providing human-understandable explanations for mutational effects and prioritizing novel mutations with desirable properties. Our code, model, and data are open-sourced at `https://github.com/PharMolix/MutaPLM`.

## 1 Introduction

Studying protein evolution through mutations within amino acid sequences is a central research topic in life sciences [1–3]. Despite immense research efforts, a large number of protein mutations with biological significance remain under-explored, highlighting the demand for in-silico tools to model these mutations. Practically, the tool should meet two requirements. First, it should be **explainable**, providing insightful and human-understandable interpretations for mutational effects. This is crucial for broad biological applications ranging from identifying immune-escape pathogens [4, 5] to interpreting the mechanisms of human diseases [6, 7]. Additionally, the tool should be **engineerable**, proposing protein mutations that satisfy desirable properties such as catalytic activity and thermostability. This process is known as directed evolution [8, 9], the most prevailing approach for protein design in the laboratory, which offers substantial benefits across various application fields, including industry [10], biotechnology [11], and therapeutics [12].

To achieve these goals, deep learning models [13–15] have emerged to capture evolutionary information from protein sequences. Recently, the development of protein language models (PLMs)

---

[†]Equal contribution
[*]Corresponding authors

38th Conference on Neural Information Processing Systems (NeurIPS 2024).

[16–20] has brought a paradigm shift in computational biology. By self-supervised learning [21] on evolutionary-scale databases [22, 23], PLMs have achieved great success in various biological applications, including structure prediction [19, 24] and protein design [18, 25]. Additionally, PLMs have demonstrated zero-shot capabilities in predicting and optimizing evolutionary plausibility [26–28], a continuous value indicating whether a mutation is favored by natural selection.

Despite their promising advancements, we argue that existing PLMs are not yet satisfactory as explainable and engineerable tools for handling protein mutations. Regarding mutation explanation, PLMs' implicit interpretation with evolutionary plausibility is overly vague, lacking detailed information for mutational effects such as specific alterations in protein functions and impacts on organisms. Regarding mutation engineering, PLMs can only propose evolutionary-plausible mutations, which may be misaligned with human preferences in real-world practices of directed evolution. For example, enhancing the catalytic activity of an enzyme from a bacterium could be detrimental to its survival due to increased energy costs but beneficial for industrial applications. In such scenarios, the utility of PLMs in assisting protein engineering is significantly compromised.

In this paper, we aim to develop explainable and engineerable PLMs by explicitly modeling protein mutations. However, conventional PLMs based on the Transformers [29] architecture provide context-aware representations for each amino acid, which are inadequate for capturing the discrepancies between the wild-type and its mutant within a unified feature space. Besides, there is a lack of supervision signals necessary for comprehending the intricate impacts of protein mutations, which require extensive background knowledge, including protein structures, protein functions, and mechanisms of biological processes.

To address these issues, we envision that (1) mutation representations could be captured from the variations of PLM representations between the wild-type and its mutant with appropriate architecture, and (2) expert-written texts from protein databases and biomedical publications provide rich cross-modal supervision for learning protein mutations. Specifically, we propose **MutaPLM**, a unified framework for interpreting and navigating **Muta**tions with **P**rotein **L**anguage **M**odels. We introduce a protein *delta* network that translates between mutations and protein *delta* features, formulating a unified feature space aligned with textual semantics. We develop a transfer learning pipeline with a chain-of-thought (CoT) strategy [30] to harvest protein mutation knowledge from biomedical texts. Additionally, we construct MutaDescribe, the first large-scale dataset containing diverse protein mutations and rich textual annotations of their effects. Using natural language as a friendly interface, the dataset facilitates the training and evaluation of mutation explanation and engineering.

Through comprehensive experiments, we demonstrate that MutaPLM is a versatile, explainable, and engineerable tool for assisting protein mutation studies. In mutation explanation, MutaPLM outperforms the strongest baseline model by 6.5% in ROUGE-L, and 19.4% of the predicted mutational effects are regarded as accurate and insightful by human experts. In mutation engineering, our model achieves an average of 0.409 recall scores on top-50 mutation proposals navigated by free-text instructions, improving ESM-2 [19] by 1.6-fold.

Our contributions are summarized as follows:

- We propose MutaPLM, a unified framework that enables protein language models to capture mutations explicitly using a protein *delta* network and cross-modal supervision.

- We build MutaDescribe, the first dataset with detailed textual annotations for protein mutations.

- We validate the effectiveness of MutaPLM in explaining and engineering protein mutations through comprehensive experiments.

## 2 Related Work

### 2.1 Protein Language Models

In analogy to large language models (LLMs) [31–34] in natural language processing (NLP), protein language models (PLMs) such as ProteinBERT [35], ProtTrans [17], ProtGPT2 [18], and ESM series [36, 19, 37] have surged in modeling protein sequences. Pre-trained by masked language modeling [38] or auto-regressive language modeling [39] on evolutionary-scale protein databases, PLMs have demonstrated outstanding predictive power on protein secondary and tertiary structures [24], protein functions [40] and protein-protein interactions [41]. More recently, explorations on PLMs unifying

**(a) Encoding Branch of the Protein *Delta* Network**

**(b) Decoding Branch of the Protein *Delta* Network**

Figure 1: **Model architecture of MutaPLM. (a) The encoding branch of the protein *delta* network.** The *delta* encoder takes the subtraction of the PLM representations of the mutant and wild-type as inputs to generate $z_\Delta$. **(b) The decoding branch of the protein *delta* network.** The key components involve a *delta* decoder that reconstructs mutant features and two prediction heads deciding the position and amino acid of the mutation.

protein sequences and natural language [42–45] have attracted rising research interest, as texts provide unstructured knowledge and a friendly user interface for studying proteins. Notably, a contemporary work [46] proposes to perform text-based protein editing by directly generating the mutated protein sequence. Unfortunately, none of the existing PLMs qualifies as an explainable and engineereable tool in modeling protein mutations, mainly owing to architectural design and lack of supervision.

## 2.2 Protein Mutation Modeling

Previous works formulate mutation explanation as learning the 'local fitness landscape', a mapping from protein sequences to specific functional activity scores [47]. Models for protein fitness prediction could be categorized as (1) alignment-based models [48, 49] trained on multiple sequence alignments (MSAs) [50], (2) PLM models [18, 19] trained on large-scale unaligned sequences, (3) inverse-folding models [27, 51] that learn protein fitness through structure-conditioned sequence distributions, and (4) hybrid models [52, 53] that combine both PLMs and MSAs. The evaluations are performed as per wild-type protein on deep mutation scanning (DMS) [54] or clinical variant [55] benchmarks. In this work, we formulate mutation explanation as a more challenging task that aims at providing textual descriptions of mutational effects for arbitrary wild-type protein and mutation.

The traditional mutation engineering [8, 9] task aims at generating protein mutants with high fitness scores. One line of work leverages generative models including variational auto-encoders (VAEs) [56], generative language models [57] and diffusion models [58] to directly generate the protein sequence conditioned on fitness scores. Another line attempts to propose mutations iteratively by greedy sampling [59], reinforcement learning [60], or proximal gradients [61] on the learned fitness landscape. Differing from prior studies, MutaPLM incorporates textual instructions instead of fitness scores as navigation and proposes mutations satisfying human preferences.

## 3 Methods

The main goal of our work is to develop explainable and engineereable PLMs by explicitly modeling protein mutations. To achieve this goal, we elaborate on the proposed MutaPLM framework, highlighting three design components: (1) a protein *delta* network that translates between mutations and protein *delta* features $z_\Delta$ (Sec. 3.1, detailed in Appendix A.1), (2) a transfer learning pipeline with a chain-of-thought strategy that harvests protein mutation knowledge from cross-modal supervision (Sec. 3.2, detailed in Appendix A.3), and (3) a specifically constructed dataset with diverse proteins and rich textual annotations of mutation effects (Sec. 3.3, detailed in Appendix B.2).

## 3.1 Protein *Delta* Network for Explicit Mutation Modeling

The protein *delta* network follows an encoder-decoder architecture, utilizing textual semantics as the latent feature space for protein mutations. As illustrated in Fig. 1, the protein *delta* network is

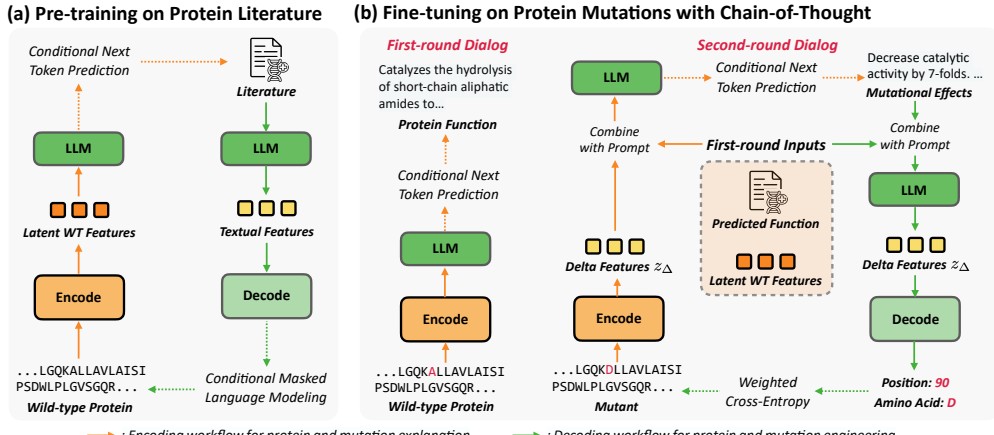

Figure 2: **Training pipeline of MutaPLM. (a) Workflow of pre-training on protein-related literature.** We perform next token prediction for the encoding workflow and conditional masked language modeling for the decoding workflow. **(b) Workflow of fine-tuning with chain-of-thought (CoT).** We employ a two-round dialog that involves describing the functions of a wild-type protein, explaining the effects of its mutation, and predicting the mutation based on the mutational effects.

composed of a protein language model (PLM), a large language model (LLM), a wild-type encoder, a *delta* encoder, a *delta* decoder, and two mutation prediction heads. We leverage ESM-2 (650M) [19], a powerful PLM pre-trained on evolutionary-scale databases, to encode protein sequences. We initialize the LLM with BioMedGPT-LM [62], a scientific language model built on LLaMA2-7B [31] through continual pre-training [63] on large-scale biomedical corpora.

**Formulation of protein *delta* features.** We speculate that the subtraction of PLM representations between the mutant and wild-type, denoted as $h_\Delta$, contains rich mutation information, making it suitable for extracting protein *delta* features $z_\Delta$. Specifically:

$$h_\Delta = h_{\text{mt}} - h_{\text{wt}} = f_{\text{PLM}}(x_{\text{mt}}) - f_{\text{PLM}}(x_{\text{wt}}), \tag{1}$$

where $x_{\text{mt}}$ and $x_{\text{wt}}$ are the amino acid sequences of the mutant and wild-type protein, $h_{\text{mt}}$ and $h_{\text{wt}}$ are their sequence representations, and $f_{\text{PLM}}$ is the protein language model.

The *delta* encoder $f_{\text{enc}}$ and *delta* decoder $f_{\text{dec}}$ facilitates bi-directional transformations between $h_\Delta$ and $z_\Delta$ as follows:

$$z_\Delta = f_{\text{enc}}(h_\Delta), \quad h_\Delta = f_{\text{dec}}(z_\Delta). \tag{2}$$

**Encoding protein *delta* features.** Given $h_\Delta$, the *delta* encoder is expected to extract information-preserving protein *delta* features $z_\Delta$ within a unified feature space. However, protein sequences vary in length, ranging from several tens to thousands of amino acids. To address this issue, we adopt a cross-attention module [29] to transform the sequential representations into a fixed number of latent features. The module, partly inspired by BLIP series [64, 65], maintains $K$ trainable features that serve as queries and takes the sequence representations as keys and values to generate outputs. We employ two parallel modules for encoding the wild-type features $h_{\text{wt}}$ and mutational features $h_\Delta$.

**Decoding protein *delta* features.** Drawing inspirations from LM-DESIGN [66], we introduce a cross-attention module that takes a symmetrical form of the *delta* encoder. Specifically, it treats the wild-type protein representations $h_{\text{wt}}$ as queries and protein *delta* features $z_\Delta$ as keys and values. The outputs are then processed by a two-layer feed-forward network (FFN) to reconstruct $h_\Delta$. The mutant representations $h_{\text{mt}}$ are obtained by combining $h_\Delta$ with $h_{\text{wt}}$, and fed into a position head and a language modeling head to predict the mutation. The position head is a fully-connected layer that predicts whether the amino acid should be substituted. The language modeling head is initialized from the PLM and predicts the type of the mutated amino acid. To facilitate text-based protein engineering, we maintain $K$ trainable soft tokens, which are appended to the input token embeddings of the LLM to summarize textual semantics. The output representations of the soft tokens are processed by the *delta* decoder to generate mutations.

Compared with previous works that connect protein sequences with LLMs [67, 44, 45], the proposed protein *delta* network exhibits the following advantages:

- Explicit modeling of protein mutations. Prior models are designed for static protein sequences, while MutaPLM models the alterations introduced by mutations with protein *delta* features $z_\Delta$.

- Encoder-decoder architecture. Prior works adopt either an encoder or a decoder architecture for protein sequences, while MutaPLM incorporates both encoding and decoding components.

## 3.2 Transfer Learning with Cross-modal Supervision

Biomedical texts contain rich expert-annotated information on protein properties and mutational effects. As depicted in Fig. 2, MutaPLM harvests these cross-modal supervision signals through a transfer learning pipeline, which we detail as follows:

**Pre-training on protein literature.** In this stage, we aim to incorporate general protein knowledge from scientific publications with language modeling objectives, as shown in Fig. 2(a). (1) For the encoding workflow, we take the output representations of the wild-type encoder as LLM inputs and calculate the next-token prediction objective [39] for generating descriptive texts. (2) For the decoding workflow, we employ the conditional masked language modeling (CMLM) objective [68] on the protein sequence. Specifically, we mask 15% amino acids and task the PLM to recover the masks based on the remaining amino acid sequence and the LLM-summarized textual representations. It is worth noting that in this stage, the *delta* decoder acts as a modality translator, generating bias terms that help reconstruct the original sequence instead of capturing protein mutation information. Overall, we optimize the summation of these two language modeling objectives.

**Fine-tuning on protein mutations with chain-of-thought (CoT).** As depicted in Fig. 2(b), we fine-tune MutaPLM on textual annotations of mutational effects to facilitate mutation explanation and engineering. Since mutational effects typically involve the enhancement or attenuation of protein functions, we adopt a chain-of-thought (CoT) strategy [30] that seamlessly connects protein functions and mutational effects within a two-round dialogue. In the first round, we prompt the LLM to describe the functions of the wild-type protein using the encoding workflow. In the second round, we introduce two tasks, namely describing the mutational effects with the encoding workflow, and predicting the mutation based on textual instructions with the decoding workflow. Both tasks utilize the latent wild-type representations and the predicted functions from the first round dialogue as additional inputs. Formally, the overall objective of fine-tuning is the summation of three parts: (1) next token prediction on protein function descriptions, (2) next token prediction on mutational effects, and (3) weighted cross-entropy between the predicted mutation and the ground-truth mutation.

Table 1: **Statistics of the MutaDescribe dataset.** We report the number of proteins and samples, the average protein sequence length, and the average number of words for mutational effects.

| Split | # Proteins | # Samples | Avg. sequence length | Avg. words |
|---|---|---|---|---|
| Train | 20,553 | 165,236 | 516.1 | 28.3 |
| Valid | 2,207 | 4,663 | 524.8 | 28.3 |
| Test (Easy) | 429 | 460 | 518.1 | 27.3 |
| Test (Medium) | 68 | 384 | 669.6 | 31.6 |
| Test (Hard) | 81 | 404 | 530.0 | 31.8 |

## 3.3 MutaDescribe: A Diverse Protein Mutation Dataset with Textual Annotations

We build MutaDescribe, a large-scale dataset comprising 20.9K wild-type proteins and 171.1K single-site mutations, to facilitate fine-tuning and evaluation. We provide an overview of our dataset in Tab. 1. The construction process involves the following steps:

**Raw data collection.** The primary source of MutaDescribe is UniProtKB/SwissProt [69], a widely adopted protein database that contains 106.6K single-site substitutions. We collect expert-reviewed descriptions of mutational effects from the *Phenotypes & Variants* entry and retrieve the abstract of the corresponding publications on PubMed [70] based on available reference information.

**Quality control.** We prompt GPT-3.5-turbo [33] to filter out low-quality descriptions such as those that only mention the originating species. This step helps ensure that the dataset contains high-quality and informative annotations.

Table 2: **Performance evaluation for mutation explanation on the test sets of MutaDescribe.** R-L: ROUGE-L. BL-2: BLEU-2.

| Model | Easy | | Medium | | Hard | | Average | |
|---|---|---|---|---|---|---|---|---|
| | R-L | BL-2 | R-L | BL-2 | R-L | BL-2 | R-L | BL-2 |
| ProLLaMA [45] | 1.02 | 0.64 | 1.00 | 0.91 | 1.03 | 0.70 | 1.02 | 0.74 |
| Mol-Instructions [67] | 5.10 | 0.65 | 5.19 | 0.65 | 5.56 | 0.90 | 5.27 | 0.73 |
| Galactica-6.7B [74] | 6.53 | 3.52 | 7.64 | 3.58 | 7.33 | 2.88 | 7.13 | 3.33 |
| GPT-4-0613 (1-shot) [33] | 8.04 | 2.93 | 9.96 | 3.42 | 9.62 | 2.69 | 9.14 | 3.00 |
| GPT-4-0613 (5-shot) [33] | 10.46 | 2.51 | 10.31 | 2.81 | 10.79 | 1.88 | 10.52 | 2.40 |
| GPT-4-0613 (5-shot, kNN) [33] | 11.63 | 9.63 | 12.98 | 10.88 | 12.46 | 8.63 | 12.31 | 9.69 |
| GPT-4 + ESM-2 [19] | 11.69 | 11.09 | 13.02 | 11.50 | 12.77 | 8.48 | 12.45 | 10.37 |
| GPT-4 + OntoProtein [75] | 11.84 | 10.93 | 12.69 | 11.22 | 12.81 | 8.17 | 12.42 | 10.13 |
| AugmentedESM [27] | 11.60 | 8.33 | 11.40 | 7.46 | 10.73 | 6.95 | 11.26 | 7.62 |
| Fine-tuned ESM-2 [19] | 20.49 | 9.37 | 11.87 | 5.95 | 11.34 | 3.32 | 14.88 | 6.36 |
| **MutaPLM** | **25.80** | **18.77** | **21.07** | **12.59** | **16.51** | **8.69** | **21.34** | **13.61** |

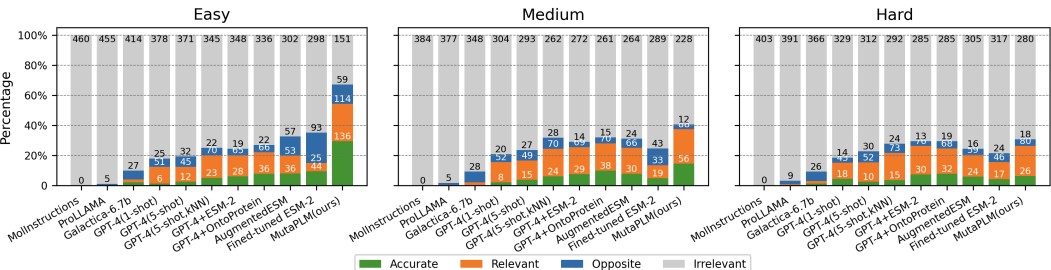

Figure 3: **Human-AI collaborative evaluation results for mutation explanation on the test sets of MutaDescribe.** We show the number of accurate, relevant, opposite, and irrelevant predictions.

**Data enrichment.** Given that the descriptions in UniProtKB are generally short and homogeneous, we utilize GPT-3.5-turbo to enrich the textual annotations by retrieving relevant descriptions from the original PubMed abstract. Additionally, we balance the number of benign and malignant mutations by constructing reversed samples. Specifically, for each mutation, we attempt to exchange the wild-type and the mutant and prompt GPT-3.5-turbo to write a description opposite to the original mutational effect. For example, if the mutational effect of an A89H mutation is *"Increased catalytic activity"*, we will create a reversed sample with an H89A mutation and *"Decreased catalytic activity"*.

**Data splitting.** We first randomly split our dataset into training, validation, and test sets. To evaluate models' generalization capabilities on novel proteins, we further partition the test set into three subsets based on the wild-type sequence homology with training sequences. We adopt MMSeqs2 [71], a widely-adopted tool to calculate sequence homology. The *Easy, Medium* and *Hard* test subsets comprise samples whose sequence homology are between $[0.95, 1]$, $[0.5, 0.95)$, and $[0, 0.5)$ respectively. We also implement a temporal split based on the publication date of the mutation, and we defer readers to Appendix B for details and Appendix D.1 for evaluation results.

Compared with prior mutation benchmarks [55, 72, 73], MutaDescribe is the first to incorporate textual annotations for facilitating mutation explanation and engineering. Besides, MutaDescribe contains a wider variety of wild-type proteins, surpassing ProteinGym [73] by 6 times in quantity.

## 4  Experiments

In this section, we demonstrate that MutaPLM is adept at interpreting and engineering mutations through comprehensive experiments. We start with a brief introduction of our training setups (Sec. 4.1), followed by detailed evaluations on two core tasks: mutation explanation (Sec. 4.2) and mutation engineering (Sec. 4.3). We also present an in-depth analysis of our design components (Sec. 4.4), including pre-training and the CoT strategy.

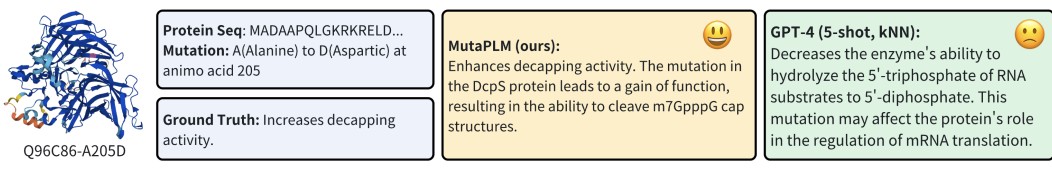

Figure 4: **Case study for a mutation from A (Alanine) to D (Aspartic) at the 205-th position of *m7GpppX diphosphatase*.** MutaPLM provides accurate explanations and insights, while GPT-4 generates irrelevant results.

## 4.1 Training Setup

To alleviate catastrophic forgetting [76] and save computational costs, we train MutaPLM in a parameter-efficient way. We apply low-rank adaptation (LoRA) [77] on the LLM with a rank of 16. The number of query embeds and soft tokens is set as $K = 32$. We optimize the LoRA modules, the wild-type encoder, the *delta* encoder, the *delta* decoder, the soft tokens, the position head, and the language modeling (LM) head, which comprises a total of 75.0M parameters. The remaining 7.4B parameters are kept frozen.

We pre-train MutaPLM for 200K steps with a batch size of 32 on 1.1M protein-text data collected from biomedical publications (detailed in Appendix B.1) and fine-tune it for 70K steps with a batch size of 24 on MutaDescribe. For both stages, we use the AdamW optimizer [78] with a learning rate that is linearly warmed up to $10^{-4}$ for the first 1K steps and decreases to $10^{-5}$ following a cosine annealing strategy. The overall training process takes 10 days on 4 NVIDIA A100 GPUs.

## 4.2 Performance Evaluation on Mutation Explanation

Differing from existing studies that interpret mutational effects with protein fitness [26, 28], we formulate mutation explanation as providing detailed textual descriptions for protein mutations.

**Baselines.** While no prior work is specifically designed for this task, we perform zero-shot analysis on popular LLMs with various zero-shot or few-shot prompts and implement supervised models for comparison. Our baselines include (1) Text-based LLMs. We perform in-context learning [79] by providing 1-shot and 5-shot demonstrations to GPT-4 [33], the most advanced model in NLP. Additionally, we implement a k-nearest neighbor (kNN) strategy [80] that selects the top-k homologous proteins from the training set as few-shot examples. (2) LLM-assisted PLMs, including ESM-2 [19] and OntoProtein [75]. In addition to kNN-based 5-shot samples for GPT-4, we leverage PLMs to provide additional information by predicting the evolutionary plausibility of the mutation. (3) LLMs trained on protein sequences, including Galactica-6.7B [74], Mol-Instructions [67], and ProtLLM [44]. We feed the wild-type and mutated protein sequences into these models and instruct them to provide mutation explanations. (4) Fine-tuned LLMs. We fine-tune BioMedGPT-LM by feeding the ESM-2 representations of the wild-type and mutant (Fine-tuned ESM-2) or the wild-type sequence and evolutionary plausibility (AugmentedESM [27]) into the LLM and performing casual generation. Notably, for all ESM-2 models used in our baselines, we adopt the model with 650M parameters for fair comparison. We defer readers to Appendix C.1 for more implementation details.

**Evaluation.** We adopt BLEU [81] and ROUGE [82] scores to assess the quality of the generations by comparing them with ground-truth annotations. To further investigate whether the predictions are truly insightful and helpful in studying protein mutations, we perform a human-AI collaborative evaluation. Specifically, we first utilize GPT-4 as a proxy of human experts to categorize the predictions into *Accurate*, *Relevant*, *Opposite*, and *Irrelevant*, based on the relevance between the predictions and ground truth. Then, we recruit a postgraduate from a top university who majors in biology to assess and rectify GPT-4 evaluation results on mutation explanations following the same categorization protocol. The prompt and detailed evaluation results are displayed in Appendix C.3.

**Results and analysis.** We present performance comparisons on the test sets of MutaDescribe in Tab. 2 and Fig. 3. We observe that: (1) MutaPLM achieves state-of-the-art performance across various evaluation metrics, outperforming fine-tuned ESM-2 by 6.46% in ROUGE-L and GPT-4 + ESM-2 by 3.24% in BLEU-2. Additionally, more than 40.22% of MutaPLM predictions are regarded as *Accurate* or *Relevant* with ground-truth labels, which showcases our model's helpfulness in real-world research scenarios. (2) While the performance on the *Medium* and *Hard* sets is not as promising as in the easy set, MutaPLM shows generalization capabilities on novel proteins, as validated by 6.44%

Table 3: **Performance evaluation for mutation engineering on the test sets of MutaDescribe.** Acc: prediction accuracy of the amino acid given the position of the mutation. Rec@50: top 50 recall of the desired mutant. -: not reported due to unaffordable computation costs (requires $\sim$ 1M forward passes).

| Model | Easy | | Medium | | Hard | | Average | |
|---|---|---|---|---|---|---|---|---|
| | Acc | Rec@50 | Acc | Rec@50 | Acc | Rec@50 | Acc | Rec@50 |
| Random | 5.23 | 0.83 | 4.94 | 0.52 | 5.20 | 1.24 | 5.13 | 0.87 |
| ProtST (ESM-2) [42] | 5.86 | - | 6.51 | - | 7.18 | - | 6.49 | - |
| GPT-4-0613 (1-shot) [33] | 10.83 | 5.00 | 10.77 | 6.92 | 12.09 | 8.79 | 11.21 | 6.81 |
| GPT-4-0613 (5-shot) [33] | 14.84 | 4.68 | 9.32 | 6.78 | 13.33 | 5.62 | 12.65 | 5.63 |
| GPT-4-0613 (5-shot, kNN) [33] | 15.97 | 7.56 | 14.29 | 7.14 | 14.77 | 7.95 | 15.06 | 7.56 |
| ESM-2 [19] | 35.21 | 23.91 | 34.63 | 22.91 | 37.87 | 28.71 | 35.84 | 25.15 |
| OntoProtein [75] | 39.78 | 28.91 | 36.45 | 26.04 | 38.61 | 29.20 | 38.37 | 28.12 |
| Fine-tuned BioMedGPT [62] | 35.21 | 7.82 | 32.29 | 5.72 | 39.60 | 12.62 | 35.73 | 8.72 |
| Fine-tuned ESM-2 [19, 83] | 52.17 | 35.65 | **52.08** | 30.60 | 50.00 | 34.65 | 51.43 | 33.77 |
| MutaPLM | **56.08** | **43.47** | 48.69 | **34.89** | **55.19** | **43.81** | **53.51** | **40.94** |

accurate and 19.80% relevant predictions on the hard set. (3) The evolutionary plausibility values are beneficial for elucidating mutational effects, as demonstrated by the slightly improved results of LLM-assisted PLMs against the plain GPT-4 counterpart. However, the superior performance of fine-tuned ESM-2 and MutaPLM indicates that integrating the mutant sequence provides richer mutational information. (4) Supervised baselines underperform few-shot GPT-4 models, especially on *Medium* and *Hard* sets and BLEU-2 scores. We observe that supervised models tend to randomly combine short textual segments from the training set, indicating overfitting problems. (5) LLMs trained on protein sequences perform poorly, as they are solely instruction-tuned on single protein sequences. Hence, we emphasize the significance of knowledge transfer from protein functions to mutational effects and their basic properties.

**Case study.** Additionally, we present a case study in Fig. 4 for a mutation from *m7GpppX diphosphatase*. Our model accurately identifies the increased decapping activity and provides novel insights beyond the ground truth. In contrast, the GPT-4 model mistakenly identifies the mutational effects as decreases in enzymic activity. More cases are available in Appendix D.3.

## 4.3 Performance Evaluation on Mutation Engineering

Differing from prior works [59–61] that perform mutation engineering with an active learning paradigm [84], we challenge models to directly propose protein mutations based on the wild-type sequence and textual instructions. As we primarily focus on single-site mutations, we formulate this as a retrieval task from $19 \times L$ possible mutants for a protein sequence of length $L$.

**Baselines.** We adopt four groups of baselines including: (1) Few-shot LLMs. Similar to mutation explanation, we prompt GPT-4 to suggest single-site mutations through in-context few-shot learning. (2) Zero-shot PLMs including ESM-2 [19] and OntoProtein [75]. We calculate the evolutionary plausibility scores following [26] for each amino acid and derive the best mutant. (3) A retrieval-based model, namely ProtST (ESM-2) [42]. We calculate the cosine similarity between PLM and textual representations of mutational effects to score and rank mutations. (4) Fine-tuned models. We fine-tune BioMedGPT [62] to directly propose a mutation based on the protein sequence and instruction. We also fine-tune ESM-2 by combining its wild-type sequence representations with BioMedBERT [83] encodings of textual instructions by a cross-attention layer. Please refer to Appendix C.1 for details of our baselines.

**Evaluation.** We report the average accuracy of the mutated amino acid on the ground-truth mutational position. We also report top-50 recall scores on all possible mutations.

**Results and analysis.** Comparisons between MutaPLM and baselines on the test sets of MutaDescribe are presented in Tab. 3. We observe that: (1) MutaPLM achieves an average of 53.51% in accuracy and 40.94% in top-50 recall, improving the original ESM-2 model by 1.6-fold. The substantial gains

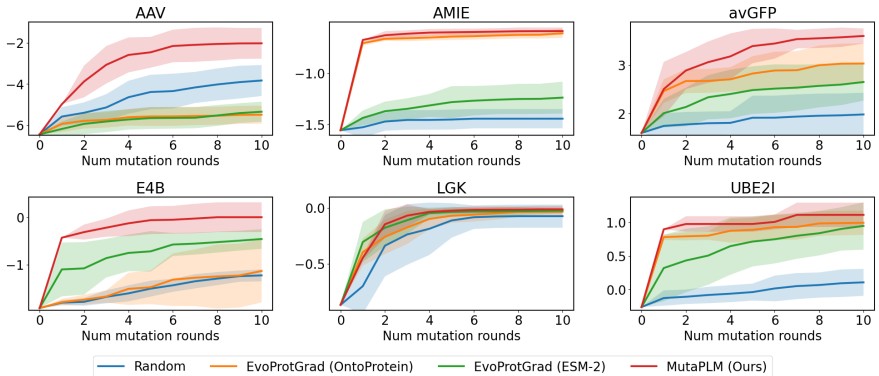

Figure 5: **Visualization of fitness scores for multi-round protein optimization.** The curves indicate the average results, and the shaded regions indicate the standard deviation.

of MutaPLM underscore the significance of textual navigation in mutation engineering. (2) MutaPLM outperforms the fine-tuned ESM-2 model by an average of 2.09% in accuracy and 6.17% in top-50 recall, benefiting from our architectural design and pre-training. (3) The overall performance of MutaPLM on the *Easy* and *Hard* sets are similar but significantly higher than on the *Medium* set. We attribute this to data distribution: protein sequences in the *Medium* set are longer (see Tab. 1), and the distribution of the mutated amino acids differs (see Fig. A1). Besides, the PLM may have witnessed the wild-type protein during pre-training, which mitigates the overfitting problem. (4) Compared to LLMs, both zero-shot and fine-tuned PLMs achieve superior performance, thanks to their evolutionary knowledge attained from pre-training on large-scale protein sequences. (5) Aligning the representations of protein sequences and texts cannot benefit mutation modeling, as evidenced by the poor performance of ProtST (ESM-2).

**Visualization of protein fitness on multi-round optimization.** In addition to single-site mutations, we employ a beam-search algorithm [85] to obtain multi-point substitutions iteratively. We manually write the optimization objective for 6 representative benchmarks, set the number of beams as 20, perform 20 independent runs, and visualize the fitness scores predicted by ESM landscape models [86]. We compare MutaPLM with EvoProtGrad [87], a gradient-based strategy that leverages PLMs for multi-round optimization, as well as with random sampling. More details are presented in Appendix C.4. As shown in Fig. 5, our model consistently yields higher-fitness mutants across 6 proteins with varying objectives, especially in the initial rounds of optimization. These results highlight MutaPLM's potential in assisting real-world mutagenesis applications.

## 4.4 In-depth Analysis

**Impacts of transfer learning.** We show the impacts of pre-training and fine-tuning in Fig. 6. As the fine-tuning proceeds, the performance of MutaPLM continues to improve on the *Easy* set but deteriorates on the *Medium* and *Hard* sets, indicating overfitting problems on out-of-domain samples. Besides, without pre-training, MutaPLM achieves higher performance for the initial steps, which we attribute to the adaptation cost from pre-training texts to fine-tuning texts. However, the overall ROUGE-L scores decline by 1.56% for mutation explanation and 1.18% for mutation engineering as the fine-tuning finalizes. Overall, these results validate our transfer learning design.

**Impacts of chain-of-thought (CoT).** To investigate the impacts of the chain-of-thought strategy, we perform ablation studies by (1) replacing the predicted function with the ground truth description, (2) replacing the predicted function with *'Unknown function'*, (3) removing the *delta* features for mutation explanation, and (4) removing the mutational effects for mutation engineering. As shown in Tab. 4, removing protein functions leads to a performance drop of 2.80% for mutation explanation and 1.13% for mutation engineering. Conversely, using the ground-truth function results in notable improvements, particularly for mutation explanation. Besides, the *delta* features and mutational effects within the second-round dialog play more significant roles in MutaPLM. These findings highlight the significance of jointly incorporating protein function and mutation information in explaining and navigating protein mutations.

Table 4: **Ablation studies.** w/o: without. w/: with. We report average ROUGE-L for mutation explanation and average Recall@50 for mutation engineering.

| Model | Explain | Engineer |
|---|---|---|
| MutaPLM | 21.34 | 40.94 |
| w/ golden function | 23.80 | 41.26 |
| w/o function | 18.54 | 39.81 |
| w/o *delta* features | 17.36 | - |
| w/o mutational effects | - | 35.10 |

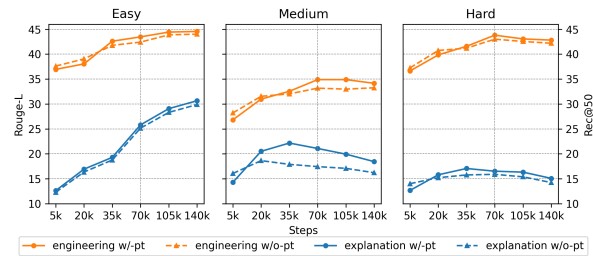

Figure 6: **Performance analysis for mutation explanation (blue) and engineering (orange) on pre-training and fine-tuning.** w/o pt: without pre-training. w/ pt: with pre-training.

# 5 Limitations and Broader Impacts

MutaPLM pioneers as the first attempt in the explicit modeling of protein mutations with natural language, and we expect future endeavors on (1) expanding the scale and diversity of the MutaDescribe dataset by integrating multi-point mutations and indels [73], (2) analyzing the alterations of protein 3D structures [88] to deepen the understanding of mutations, and (3) developing active learning [84] pipelines to harness feedbacks from wet-lab experiments in real-world mutagenesis studies.

While MutaPLM bears promise in mutation explanation and engineering, we emphasize safety concerns that it can be misused to generate pathogenic mutations and harmful bio-agents. Hence, we declare that MutaPLM, upon public release, should be restricted to research purposes, and any further applications should undergo comprehensive experiments and human inspections.

# 6 Conclusions

In this work, we present MutaPLM, a unified framework harvesting protein language models for mutation explanation and engineering. We propose a protein *delta* network to model mutations explicitly with protein *delta* features and develop a transfer learning pipeline with a chain-of-thought strategy to integrate protein mutation knowledge from biomedical texts. Additionally, we construct MutaDescribe, the first large-scale dataset containing diverse proteins and detailed textual annotations for mutations. Our experiments demonstrate that MutaPLM offers insightful explanations for mutational effects and proposes desirable mutants based on textual instructions. We anticipate that the proposed MutaPLM framework and our publicly released dataset will pave the way for novel research avenues and applications in studying proteins.

## Acknowledgments and Disclosure of Funding

This research is supported by the National Key R&D Program of China (No. 2022YFF1203002) and PharMolix Inc.

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

# Appendix

## A Details of MutaPLM

### A.1 Model Architecture

Our protein *delta* network consists of a protein language model (PLM), a large language model (LLM), a wild-type encoder, a *delta* encoder, a *delta* decoder, and two prediction heads for mutation. We introduce these components as follows:

**Protein language model.** We formulate the wild-type protein as an amino acid sequence $x_{\text{wt}} = \left[ x_1^{(\text{wt})}, x_2^{(\text{wt})}, \cdots, x_L^{(\text{wt})} \right]$ of length $L$. We focus on single-site substitution mutants, denoted by its sequence $x_{\text{mt}} = \left[ x_1^{(\text{mt})}, x_2^{(\text{mt})}, \cdots, x_L^{(\text{mt})} \right]$ satisfying $\mathcal{H}(x_{\text{wt}}, x_{\text{mt}}) = 1$, where $\mathcal{H}(\cdot, \cdot)$ is the Hamming distance. We adopt ESM-2 (650M) [19] as our protein language model $f_{\text{PLM}}$, which transforms the protein sequences into dense feature vectors as follows:

$$
\begin{aligned}
h_{\text{wt}} &= \left[ h_1^{(\text{wt})}, h_2^{(\text{wt})}, \cdots, h_L^{(\text{wt})} \right] = f_{\text{PLM}}(x_{\text{wt}}), \\
h_{\text{mt}} &= \left[ h_1^{(\text{mt})}, h_2^{(\text{mt})}, \cdots, h_L^{(\text{mt})} \right] = f_{\text{PLM}}(x_{\text{mt}}).
\end{aligned}
\tag{A1}
$$

Then, we introduce the mutational representation, $h_\Delta$, calculated as follows:

$$
h_\Delta = \left[ h_1^{(\Delta)}, h_2^{(\Delta)}, \cdots, h_L^{(\Delta)} \right] = h_{\text{mt}} - h_{\text{wt}}.
\tag{A2}
$$

**Large language model.** Similarly, we formulate biomedical texts as a sequence of tokens $t = [t_1, t_2, \cdots, t_N]$. We initialize our LLM with BioMedGPT-LM [62], which is obtained by continually pre-training LLaMA2-7B [31] on biomedical corpus. The large language model $f_{\text{LLM}}$ takes the following steps to transform $t$ into latent features and output distributions of the next token:

$$
\begin{aligned}
e &= [e_1, e_2, \cdots, e_N] = g_{\text{emb}}(t), \\
z_t &= [z_1, z_2, \cdots, z_N] = g_{\text{transformers}}(e), \\
P(t_i|t_{<i}) &= g_{\text{LM}}(h_i),
\end{aligned}
\tag{A3}
$$

where $g_{\text{emb}}$ is the word embedding layer, $z_t$ is the textual representation calculated by transformer blocks $g_{\text{transformers}}$, $g_{\text{LM}}$ is the language modeling head, and $P(t_i|t_{<i})$ is the probability distribution of $i$-th token based on preceding tokens.

**Wild-type encoder.** The wild-type encoder comprises $K$ trainable query vectors $q_{\text{wt}} = [q_1, q_2, \cdots, q_K]$ and a cross attention module. It transforms the wild-type representations $h_{\text{wt}}$ into a fixed number of features as follows:

$$
z_{\text{wt}} = \left[ z_1^{(\text{wt})}, z_2^{(\text{wt})}, \cdots, z_K^{(\text{wt})} \right] = \text{CrossAttention}_{\text{wt}}(q_{\text{wt}}, h_{\text{wt}}, h_{\text{wt}}),
$$

$$
\text{CrossAttention}(Q, K, V) = \text{Softmax}\left( \frac{\hat{Q}\hat{K}^T}{\sqrt{d_k}} \right) \hat{V}
\tag{A4}
$$

$$
\hat{Q} = QW^Q, \hat{K} = KW^K, \hat{V} = VW^V,
$$

where $W^Q, W^K, W^V$ are trainable parameters, and $d_k$ is the feature dimension.

***Delta* encoder.** The *delta* encoder follows the same architecture as the wild-type encoder. It encodes the protein *delta* features as follows:

$$
z_\Delta = \left[ z_1^{(\Delta)}, z_2^{(\Delta)}, \cdots, z_K^{(\Delta)} \right] = \text{CrossAttention}_{\text{enc}}(q_\Delta, h_\Delta, h_\Delta),
\tag{A5}
$$

where $q_\Delta$ are the $K$ trainable queries, and the cross attention is calculated following Equ. A4. Notably, the wild-type encoder and *delta* encoder comprise independent parameters.

***Delta* decoder.** The *delta* decoder transforms the protein *delta* features $z_\Delta$ back to the original mutation representations $h_\Delta$. It comprises a cross-attention layer and a two-layer feed-forward network with ReLU activation. Specifically:

$$
\begin{aligned}
\tilde{z}_\Delta &= \text{CrossAttention}_{\text{dec}}(h_{\text{wt}}, z_\Delta, z_\Delta), \\
h_\Delta &= \text{FeedForward}(\tilde{z}_\Delta),
\end{aligned}
\tag{A6}
$$

Table A1: **Average $l_2$-norm of MutaPLM's intermediate representations on MutaDescribe.**

| Representation | $h_{\text{wt}}$ | $h_\Delta$ | $z_\Delta$ |
|---|---|---|---|
| Avg. $l_2$-norm | 9.90 | 0.35 | 1.04 |

where the cross attention is calculated following Equ. A4.

**Mutation prediction heads.** After reconstructing the mutant representation by $h_{\text{mt}} = h_{\text{wt}} + h_\Delta$, we develop a position prediction head $f_{\text{pos}}$ and a language modeling head $f_{\text{LM}}$ to predict the mutation. Specifically:

$$
\begin{aligned}
P\left(x_i^{(\text{mt})} \neq x_i^{(\text{wt})}\right) &= f_{\text{pos}}\left(h_i^{(\text{mt})}\right), \\
P\left(x_i^{(\text{mt})}\right) &= f_{\text{LM}}\left(h_i^{(\text{mt})}\right),
\end{aligned}
\tag{A7}
$$

where $P\left(x_i^{(\text{mt})} \neq x_i^{(\text{wt})}\right)$ denotes the probability of $i$-th amino acid to be mutated, and $P\left(x_i^{(\text{mt})}\right)$ denotes the probability distribution of the $i$-th amino acid. The parameters of the position prediction head are initialized from scratch, and those of the language modeling head are derived from the PLM.

## A.2 Justifications for Mutational Features

To model mutations explicitly, we leverage the subtraction of the wild-type and mutant representations as the mutational features $h_\Delta$, which is subsequently processed by the *delta* encoder. One of the essential considerations is that the PLM is overly smooth, making $h_\Delta$ too small and less informative. However, we argue that due to the non-smooth nature of the protein fitness landscape [61], the output representations of PLMs are also non-smooth. Moreover, after training, the *delta* encoder learns to capture the orientation of $h_\Delta$, yielding a $z_\Delta$ with an appropriate norm. We also present empirical justification by calculating the average $l_2$-norm of $h_{\text{wt}}$, $h_\Delta$, and $z_\Delta$ on MutaDescribe, which are displayed in Tab. A1.

## A.3 Pre-training Objectives

MutaPLM performs pre-training on large-scale protein-relevant literature. Given the protein sequence $x_{\text{wt}}$ and its semantically related text $t$, we optimize the following objectives:

**Protein-to-text generation.** We first concatenate the latent wild-type features $z_{\text{wt}}$ in Equ. A4 and the text embeddings $e$ in Equ. A3. We perform conditional auto-regressive language modeling that aims to generate $t$ based on the protein representations and previous tokens. The objective is calculated as follows:

$$
\begin{aligned}
z &= [\underbrace{z_1, z_2, \cdots, z_K}_{\text{protein}}, \underbrace{z_{K+1}, \cdots, z_{K+N}}_{\text{text}}] = g_{\text{transformers}}\left([z_{\text{wt}}; e]\right), \\
P(t_i|t_{<i}, z_{\text{wt}}) &= g_{\text{LM}}(z_{K+i}), \\
\mathcal{L}_{p2t} &= \frac{1}{N}\sum_{i=1}^{N} H\left[t_i, P(t_i|t_{<i}, z_{\text{wt}})\right],
\end{aligned}
\tag{A8}
$$

where $H(\cdot, \cdot)$ denotes cross-entropy.

**Text-to-protein generation.** We first append $K$ trainable soft tokens $s = [s_1, s_2, \cdots, s_K]$ to the input token embeddings to summarize textual semantics. Then, we derive $z_\Delta$ as the last hidden state of $s$ as follows:

$$
\tilde{z} = [\underbrace{\tilde{z}_1, \tilde{z}_2, \cdots, \tilde{z}_N}_{\text{text}}, \underbrace{\tilde{z}_{N+1}, \cdots, \tilde{z}_{N+K}}_{z_\Delta}] = g_{\text{transformers}}([e; s]),
\tag{A9}
$$

where $s$ denotes the soft tokens. We pass $z_\Delta$ into the *delta* decoder to obtain $h_\Delta$ as in Equ. A6. It is worth noting that in this stage, we are aimed at aligning the feature space of PLMs and LLMs, and $z_\Delta$ and $h_\Delta$ are **NOT** related to protein mutations.

Then, we randomly mask 15% amino acids in the protein sequence. We adopt the conditional masked language modeling objective to reconstruct the masked tokens as follows:

Table A2: **Prompt templates for fine-tuning.** The first and second round dialogs are composed of system prompts, latent wild-type and *delta* features, and special tokens including `<BOP>`, `<EOP>`, `<BOM>`, `<EOM>`. We highlight the parts that are used to calculate the objectives.

| Type | Content |
|---|---|
| System Prompt | You are an expert at biology and life science. Now a user gives you several protein sequences and mutations. Please follow user instructions and answer their questions. |
| User Prompt for Function Prediction | Based on the following protein sequence, please describe its function. |
| User Prompt for Mutation Explanation | Next is a mutation from `<`$x_i$`>` to `<`$\hat{x}_i$`>` at position $i$. Please generate a brief/detailed introduction to describe it. |
| User Prompt for Mutation Engineering | Next is a brief/detailed introduction of mutational effects. Please generate a protein mutation that fits the description. |
| Round 1 Dialog | `[System Prompt] [User Prompt for Function Prediction] <BOP>` `[Latent Wild-type Features] <EOP>` `[Protein Function]` |
| Round 2 Dialog for Mutation Explanation | `[Round 1 Dialog] [User Prompt for Mutation Explanation]` `<BOM>` `[`*Delta* `Features] <EOM>` `[Mutational Effects]` |
| Round 2 Dialog for Mutation Engineering | `[Round 1 Dialog] [User Prompt for Mutation Engineering]` `[Mutational Effects] <BOM>` `[Soft Embeds] <EOM>` |

$$h_{\text{mask}} = f_{\text{PLM}}(x_{\text{mask}}),$$
$$\tilde{h} = \left[\tilde{h}_1, \tilde{h}_2, \cdots, \tilde{h}_L\right] = h_{\text{mask}} + h_\Delta,$$
$$P\left(x_i^{(\text{wt})}\big|x_{\text{mask}}, h_\Delta\right) = f_{\text{LM}}\left(\tilde{h}_i\right), \tag{A10}$$
$$\mathcal{L}_{t2p} = \frac{1}{|\mathcal{M}|}\sum_{i \in \mathcal{M}} H\left[x_i, P\left(x_i^{(\text{wt})}\big|x_{\text{mask}}, h_\Delta\right)\right],$$

where $x_{\text{mask}}$ is the masked sequence of the wild-type $x_{\text{wt}}$, and $\mathcal{M}$ denotes the masked positions.

**Overall objective.** The overall objective for pre-training is calculated by:

$$\mathcal{L}_1 = \mathbb{E}_{(x_{\text{wt}}, t) \sim \mathcal{D}_1}(\mathcal{L}_{p2t} + \mathcal{L}_{t2p}), \tag{A11}$$

where $\mathbb{E}$ denotes expectation, and $\mathcal{D}_1$ denotes our pre-training dataset.

## A.4 Fine-tuning Objectives

We employ a chain-of-thought (CoT) strategy to reason over protein functions and mutational effects in a two-round dialog. Given the wild type sequence $x_{\text{wt}}$, the mutant sequence $x_{\text{mt}}$, the description of protein functions $t_{\text{func}}$ and the description of mutation effects $t_\Delta$, we calculate the following objectives:

**First-round dialog.** We first prompt the LLM to generate function descriptions $t_{\text{func}} = \left[t_1^{(\text{func})}, t_2^{(\text{func})}, \cdots, t_M^{(\text{func})}\right]$ based on the wild-type protein. We perform conditional auto-regressive language modeling as follows:

$$\mathcal{L}_{\text{func}} = \frac{1}{M}\sum_{i=1}^{M} H\left[t_i^{(\text{func})}, P\left(t_i^{(\text{func})}\big|t_{<i}^{(\text{func})}, z_{\text{wt}}\right)\right]. \tag{A12}$$

The predictions of protein functions $y_{\text{func}} = \left[y_1^{(\text{func})}, y_2^{(\text{func})}, \cdots, y_N^{(\text{func})}\right]$ is derived by:

$$y_i^{(\text{func})} = \text{argmax}\left\{P\left(y_i^{(\text{func})}\big|y_{<i}^{(\text{func})}, z_{\text{wt}}\right)\right\} \tag{A13}$$

**Second-round dialog for mutation explanation.** We prompt the LLM to generate textual descriptions for mutation effects $t_\Delta = \left[t_1^{(\Delta)}, t_2^{(\Delta)}, \cdots, t_T^{(\Delta)}\right]$ based on the function information in the

first-round dialog and protein *delta* features $z_\Delta$. The objective is calculated as follows:

$$\mathcal{L}_{\text{exp}} = \frac{1}{T} \sum_{i=1}^{T} H \left[ t_i^{(\Delta)}, P\left( t_i^{(\Delta)} | t_{<i}^{(\Delta)}, y_{\text{func}}, z_\Delta, z_{\text{wt}} \right) \right]. \tag{A14}$$

**Second-round dialog for mutation engineering.** We apply the same soft tokens $s$ as in pre-training to the input prompt to calculate the *delta* features based on the first-round dialog and descriptions of mutational effects:

$$\hat{z} = [\underbrace{\hat{z}_1, \hat{z}_2, \cdots, \hat{z}_N}_{\text{prompt}}, \underbrace{\hat{z}_{N+1}, \cdots, \hat{z}_{N+K}}_{z_\Delta}] = g_{\text{transformers}}([t_{\text{prompt}}; s]), \tag{A15}$$

where $t_{\text{prompt}}$ is the input embeddings of the prompt involving the first-round dialog and the mutational effects.

Then, reconstructing $h_{\text{mt}} = h_{\text{wt}} + h_\Delta$ with the *delta* decoder, we calculate the weighted cross-entropy loss for the mutation position and the mutated amino acid with the prediction heads:

$$\begin{aligned}
\mathcal{L}_{\text{eng}} = -\frac{1}{L} \sum_{i=1}^{L} \Big\{ & \mathbb{1}\left\{ x_i^{(\text{mt})} = x_i^{(\text{wt})} \right\} \log(1 - f_{\text{pos}}(h_i^{\text{mt}})) \\
& + \lambda \cdot \mathbb{1}\left\{ x_i^{(\text{mt})} \neq x_i^{(\text{wt})} \right\} \log f_{\text{pos}}(h_i^{\text{mt}}) \\
& - L \cdot \mathbb{1}\left\{ x_i^{(\text{mt})} \neq x_i^{(\text{wt})} \right\} H \left[ x_i^{(\text{mt})}, f_{\text{LM}}(h_i^{(\text{mt})}) \right] \Big\},
\end{aligned} \tag{A16}$$

where $\mathbb{1}\{\cdot\}$ is the boolean indicator function, and $\lambda$ is a hyper-parameter controlling label weight. In our experiments, we set $\lambda = 50$.

The overall objective is calculated as follows:

$$\mathcal{L}_2 = \mathbb{E}_{(x_{\text{wt}}, x_{\text{mt}}, t_{\text{func}}, t_\Delta) \sim \mathcal{D}_2} (\mathcal{L}_{\text{func}} + \mathcal{L}_{\text{exp}} + \mathcal{L}_{\text{eng}}), \tag{A17}$$

where $\mathcal{D}_2$ is our fine-tuning dataset.

The prompt templates for fine-tuning are displayed in Tab. A2.

## B  Training data

### B.1  Pre-training Data

Our pre-training data involves 1.1M protein-text pairs collected from the UniProtKB/SwissProt [69] database. We download 467.8K proteins with the *Publications* entry and retrieve 257.2K PubMed [70] abstracts based on the reference information.

### B.2  Fine-tuning and Testing Data: MutaDescribe

To create a natural language annotated dataset for protein mutations, we first collect 164K samples from the *Phenotypes & Variants* entry of UniProtKB/SwissProt. After deduplication and removing sites without valid text annotations, we obtain 107K mutants for 21K proteins as our raw data, comprising 33K natural variants and 74K mutagenesis sequences.

Unfortunately, the collected raw data is not suitable for protein mutation modeling, mainly owing to the following problems: (1) As shown in Tab. A3, the expert-revised annotations within UniProtKB contain an average of 9.4 words, containing limited information. (2) Through analyzing the polarity of the mutational effects, we observe that the number of malignant and benign mutations are imbalanced ($\sim$ 9:1), which may mislead model predictions.

To address these issues, (1) we perform data enrichment by collecting the abstracts of the biological literature in which the mutation is mentioned. We retrieve 50K publications based on the reference information of the mutation available in UniProtKB and prompt GPT-3.5-turbo to extract relevant information from the abstracts. The prompt template is visualized in Tab. A5. After ChatGPT enrichment, the textual annotations are expanded with an average of 28.3 words. (2) We generate 64.5K additional reverse samples. Specifically, for each malignant and benign mutation, we exchange

Table A3: **An Overview of MutaDescribe.**

| # All | # Raw | # Enriched | # Reversed |
|---|---|---|---|
| 171,147 | 106,645 | 57,147 | 64,502 |

| Avg. words (UniProtKB) | | Avg. words (Enriched) | |
|---|---|---|---|
| 9.44 | | 28.33 | |

| # Malignant | # Benign | # Not significant | # Unknown |
|---|---|---|---|
| 72,198 | 8,000 | 26,447 | 4 |

the wild-type and mutant and prompt GPT-3.5-turbo to flip the polarity of the textual descriptions for mutational effects. We empirically find that the quality of mutation descriptions using GPT-3.5-turbo and GPT-4 is similar, and therefore we opt for GPT-3.5-turbo to save API costs.

We implement two splitting strategies for our dataset. For **structural split**, we first partition our dataset into training, validation, and test sets. Then, for each wild-type sequence in the test set, we calculate the maximum sequence homology with the wild-type sequences in the training set by MMseqs2 [71]. Based on the homology, we divide the test set into three subsets. The *Easy* subset comprises 460 mutants with homology between 0.95 and 1, the *Medium* subset comprises 384 mutants with homology between 0.5 and 0.95, and the *Hard* subset comprises 404 mutants with homology between 0 and 0.5. For **temporal split**, we extract the publication date of the literature reporting each mutation. Mutations studied before 2022 are used as training and validation sets, while those studied in 2022 and 2023 comprise the test set. The train/valid/test set comprises 156K, 8K, and 1.6K samples, respectively. The detailed statistics of temporal split are shown in Tab. A4.

We present a closer look at our MutaDescribe dataset in Fig. A1, displaying the length of protein sequences, the number of words in textual annotations, the number of mutation samples per protein, the distribution of the originating species, the distribution of the cellular localization and the distribution of the mutated amino acid. We show in our illustrations that MutaDescribe is a large-scale, diverse, and detailed annotated dataset for studying protein mutations.

## C Experiment Settings

### C.1 Baselines for Mutation Explanation

For mutation explanation, we implement the following baselines:

**Galactica-6.7B [74].** This baseline is a unified large-language model pre-trained on scientific papers and protein knowledge bases. We prompt the model to investigate if it could explain mutational effects in a zero-shot manner.

**ProLLaMA [45].** This baseline is developed on LLaMA2-7B by further pre-training the model on protein sequences from UniRef50 [22]. Similarly, we perform zero-shot mutation explanation by prompting.

**Mol-Instructions [67].** We implement the protein-oriented model of Mol-Instructions that is instruction-tuned from LLaMA2-7B [31]. We perform zero-shot prompting that provides the model with the name and amino acid sequence of the protein sequence and task definitions.

**GPT-4 [33] with in-context learning.** We adopt the 0613 version of GPT-4, the most advanced LLM in natural language processing. In addition to the protein name, wild-type sequence, and mutation

Table A4: **Statistics of the temporal split.** We report the number of proteins and samples, the average protein sequence length, and the average number of words for mutational effects.

| Split | # Proteins | # Samples | Avg. sequence length | Avg. words |
|---|---|---|---|---|
| Train | 20,295 | 156,300 | 518.00 | 28.48 |
| Valid | 5,436 | 8,000 | 514.30 | 28.73 |
| Test | 310 | 1,611 | 536.67 | 26.37 |

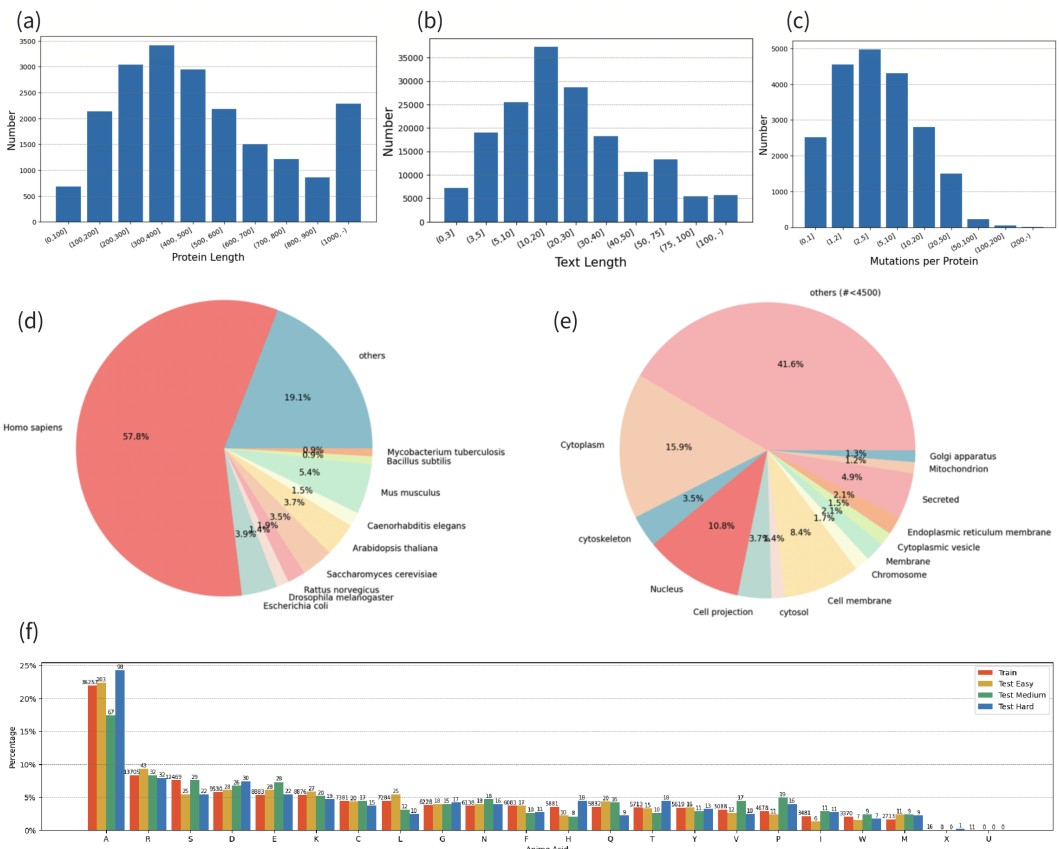

Figure A1: **Detailed statistics of the MutaDescribe dataset.** We show (a) the length of protein sequences, (b) the number of words in textual annotations, (c) the number of mutation samples per protein, (d) the distribution of the originating species, (e) the distribution of the cellular localization and (f) the distribution of the mutated amino acid.

Table A5: **Prompt template for data enrichment.** We prompt GPT-3.5-turbo to extract relevant information from the abstracts of the biological literature in which the mutation is mentioned.

[System prompt] You will be provided with a document and some relevant mutation sites (for example, site A21D indicates a mutation from A to D at position 21). First, determine whether these sites are mentioned in the document. If so, extract the text from the document that describes the functional changes caused by these sites. Otherwise, you must extract any functional changes mentioned in the document. For each site, please try to extract the corresponding protein name or gene name. You must be accurate and clear. Return a series of JSON documents, with each JSON formatted as follows:
{"Mutation Site": <provided mutation site>,
"Mentioned": <whether this site is mentioned in the document>
"protein_name": <protein name corresponding to the site>,
"gene_name": <gene name corresponding to the site>,
"Functional_changes": <functional information>}

[User prompt]
document: <document>
sites: <list of mutation sites>

Table A6: **Prompt templates for each baseline for mutation explanation.** {original_aa} and {mutated_aa} denote the amino acid before and after the mutation respectively. {fitness_change} is the subtraction of the PLM-calculated evolutionary plausibility scores between the mutant and wild-type.

| Baseline | Prompt |
|---|---|
| Galactica-6.7B | protein {protein_name}: [START_AMINO] {protein_sequence} [END_AMINO] has a mutation {original_aa} to {mutated_aa} at position {position}. Question: What are the functional changes of the protein after this mutation?
Answer: |
| ProLLaMA | Seq=<{protein_sequence}> has a mutation {original_aa} to {mutated_aa} at position {position}. Question: What are the functional changes of the protein after this mutation?
Answer: |
| Mol-Instructions | Please evaluate protein {protein_name} with the given mutation, and provide an explanation of any activity or reaction the mutation may cause:
<protein> "'{protein_sequence}"'
<mutation> {original_aa} to {mutated_aa} at position {position} |
| GPT-4-0613 (few-shot) | [System prompt] You are an expert in bioinformatics. You will be provided with a protein and its mutation information. Please predict the changes in the protein's function after this mutation. Your response should only focus on the effect of the change without additional words.

[User prompt] Example 1:
protein name: Glutathione S-transferase P
protein sequence: MPPYTVVYFPVRGRCAALRM...
mutation: D to A at position 99
(Additional samples ...)

protein name: {protein_name}
protein sequence: {protein_sequence}
mutation: {original_aa} to {mutated_aa} at position {position}
function change: |
| GPT-4 + ESM-2 &
GPT-4 + OntoProtein | [System prompt] You are an expert in bioinformatics. You will be provided with a protein and its fitness score after a single mutation. Please predict the changes in the protein's function based on the fitness score. Your response should only focus on the effect of the change without additional words.

[User prompt] Example 1:
protein name: Glutathione S-transferase P
protein sequence: MPPYTVVYFPVRGRCAALRM...
mutation: D to A at position 99
fitness change: -0.7684
(Additional samples ...)

protein name: {protein_name}
protein sequence: {protein_sequence}
mutation: {original_aa} to {mutated_aa} at position {position}
fitness change: {fitness_change}
function change: |

information, we provide few-shot demonstrations to facilitate in-context learning. For the 1-shot and 5-shot baseline, we randomly sample 1 and 5 samples from the training set of MutaDescribe. For the kNN-based 5-shot baseline, we follow [80] to search for relevant samples based on the sequence homology calculated by MMseqs2 [71]. We select 5 samples from the training set with the highest homology as few-shot demonstrations for each test sample.

**GPT-4 + ESM-2 [19].** ESM-2 is a popular protein language model pre-trained on evolutionary-scale databases. Given a mutation, we mask the mutated position and utilize ESM-2 (650M) to predict the logits for the mutated amino acid. Following [26], we adopt the subtraction between the mutant and wild-type logits as the evolutionary plausibility scores. We follow the 5-shot kNN setting on GPT-4 and provide the scores as additional information.

**GPT-4 + OntoProtein [75].** OntoProtein is a text-augmented PLM that aligns protein sequences with gene ontology definitions. We follow the GPT-4 + ESM-2 baseline to predict mutational effects based on evolutionary plausibility and kNN few-shot demonstrations.

**AugmentedESM [27].** In the original paper, the model is designed to solve fitness regression tasks by linearly combining the adaptive fitness score calculated following [26] and the amino acid sequence. We slightly adapt the model to perform mutation explanation by feeding the fitness score and the raw protein sequence into BioMedGPT-LM. We fine-tune the LLM with the casual auto-regressive language modeling objective on mutation effects. The hyperparameters for fine-tuning are the same as MutaPLM.

**Finetuned ESM-2.** Similar to MiniGPT-4 [89], we translate each residue representation of ESM-2 (650M) [19] into LLM input embeddings using a linear projection layer. We fine-tune BioMedGPT-LM with the casual auto-regressive language modeling objective on mutation effects based on the translated features of the wild-type and mutant. The hyperparameters for fine-tuning are also the same as MutaPLM.

The prompts for our baselines are displayed in Tab. A6.

## C.2   Baselines for Mutation Engineering

For mutation engineering, we implement the following baselines:

**Random.** As the name suggests, the proposed mutations are randomly sampled from every possible single-site substitution with equal probability.

**GPT-4 [33] with in-context learning.** We provide few-shot examples for GPT-4 to suggest protein mutations, and the sampling strategy is the same as in mutation explanation. We evaluate accuracy and top-50 recall with a two-round dialog. In the first-round dialog, we directly prompt GPT-4 to provide 50 mutations on arbitrary positions. In the second-round dialog, we provide the model with the ground-truth position and ask

**ESM-2 [19].** We feed the whole sequence into the PLM to calculate the output logits for each amino acid. We rank mutations by the subtraction of the mutant and wild-type logits.

**OntoProtein [75].** This baseline follows the same implementation as ESM2-650M.

**ProtST (ESM-2) [42].** ProtST trains a series of PLMs by contrastive learning [90] between protein sequences and biomedical texts. Hence, we implement a cross-modal retrieval strategy, using the cosine similarity between the mutated sequence and the textual description of mutational effects to score mutations. We opt not to report top-50 recall scores due to: (1) unaffordable computational costs, as each possible mutation requires an individual forward pass, and (2) poor performance, as the baseline merely outperforms random guesses.

**Fine-tuned BioMedGPT.** We provide the LLM with the wild-type sequence and textual instructions of desired mutational effects, and fine-tune the model to propose mutations. To evaluate accuracy, we additionally provide the mutated position and prompt the model to generate the mutated amino acid. To evaluate top-50 recall, we prompt the model to generate a single mutation, since our dataset only comprises one ground-truth mutation. The evaluations are performed within two independent sessions, and we combine the causal auto-regressive language modeling objective of both sessions during fine-tuning.

Table A7: **Prompt template for few-shot GPT-4 and fine-tuned BioMedGPT in mutation engineering.**

| Evaluating Rec@50 on GPT-4 | [System prompt] You are an expert in bioinformatics. You will be provided with a protein and the functional change resulting from a single-site mutation. Please predict the 50 most probable mutation sites where Each entry starts with the amino acid before the mutation, followed by the position of the mutation, and ends with the amino acid after the mutation. For example, D65A indicates that the amino acid at position 65 changes from D to A. Your response should only contain the 50 sites in a list format separated by commas, without additional words. |
|---|---|
| | [User prompt] Example 1:
protein name: Glutathione S-transferase P
sequence: MPPYTVVYFPVRGRCAALRMLLA...
functional change: Reduces affinity for glutathione.
50 probable mutation sites: D99A, T110K, D58V, L53I, V165P, ...
(Additional samples ...)

protein name: {protein name}
sequence: {protein sequence}
functional change: {mutational effects}
50 probable mutation sites: |
| Evaluating Accuracy on GPT-4 | {First round dialog}
[User prompt] The correct mutated position is {mutation position}. What is the most probable amino acid after the mutation? The valid amino acids include: [G, V, S, E, C, K, Q, N, M, H, I, Y, L, D, W, A, T, R, P, F]. Your answer should only contain one of the uppercase amino acids without other words. |
| Evaluating Rec@50 on fine-tuned BioMedGPT | You are an expert assistant in biology and protein engineering. Now you are given a protein sequence and an instruction describing a mutation effect.

Protein: {protein sequence}
Instruction: {mutational effects}
User: Please design a mutation that best fits the instruction.
Assistant: |
| Evaluating Accuracy on fine-tuned BioMedGPT | You are an expert assistant in biology and protein engineering. Now you are given a protein sequence and an instruction describing a mutation effect.

Protein: {protein sequence}
Instruction: {mutational effects}
User: Given mutation at position {mutation position}, please choose an amino acid that best fits the instruction.
Assistant: |

**Fine-tuned ESM-2.** We leverage BioMedBERT [83] to encode the textual instructions. We employ a cross-attention layer that takes the ESM-2 representations of the wild-type sequence as queries and the BioMedBERT representations as keys and values. The outputs are fed into a position prediction head and a language modeling head to predict mutations, which is the same as MutaPLM.

The prompt templates for GPT-4 and fine-tuned BioMedGPT are presented in Tab. A7.

### C.3   Human-AI Collaborative Evaluation for Mutation Explanation

Due to the complexity of biomedical texts, we develop a human-AI collaborative evaluation pipeline to comment on the accuracy and helpfulness of predicted mutational effects. Specifically, we query GPT-4 to compare model predictions with ground-truth annotations as in Tab. A8 and categorize them as follows.

Table A8: **Prompt template for GPT-4 evaluation.** We leverage GPT-4 to categorize predictions into *Accurate, Relevant, Opposite*, and *Irrelevant*, based on the relevance between the predicted functional alterations and ground-truth explanations.

[`System prompt`] You are an expert in biology and protein sciences. You want to figure out the effects of protein mutations by alterations of protein functions. Now we provide you with two descriptions of protein mutational effects in a JSON format, where the "label" denotes the ground truth description of the mutational effects, and the "prediction" denotes the prediction of a model. You should be precise and faithful in evaluating if the predicted mutation effects are semantically related to the ground truth. You should answer with one of the following categories:

(1) Accurate. The prediction and the label describe the same functions that are altered, and the extent of functional changes is mostly the same (For example, "strongly decrease" and "abolish").
(2) Relevant. The prediction and the label describe the same functions that are altered, and the extent of functional changes are in the same direction (For example, "strongly increase" and "slightly increase").
(3) Opposite. The prediction and the label describe the same functions that are altered, but the functional changes are opposite (For example, "increase" and "decrease").
(4) Irrelevant. The prediction and the label describe different alterations of functions.
Note that you should be careful about the altered functions before analyzing the extent. Answer with one word only from "Accurate", "Relevant", "Opposite" and "Irrelevant" to summarize your evaluation.

[`User prompt`]{"label": {ground_truth}, "prediction": {model_output}}

---

- *Accurate.* The predicted alterations in protein functions and estimations of extent are mostly the same as the ground truth.

- *Relevant.* The prediction identifies the protein function that is altered by the mutation. While it accurately predicts the attenuation or the degradation, the estimation of the extent is not correct.

- *Opposite.* The prediction identifies the protein function that is altered by the mutation. However, it mistakenly predicts attenuation as degradation or vice versa.

- *Irrelevant.* The prediction and the ground truth are about completely different functional alterations.

Then, we recruit a postgraduate from a top university who majors in biology to further assess the results. Specifically, we collect samples that are marked as *Accurate*, *Relevant*, and *Opposite* by GPT-4, and include *Irrelevant* samples for strong baselines (5-shot GPT-4 models and fine-tuned models) and MutaPLM. We present the mutation explanations, ground-truth results, GPT-4 evaluation, and categorization protocol, and ask the expert to rectify the evaluation result if necessary. In total, 12.0% of the GPT-4 evaluations are modified, and the confusion matrix is displayed in Fig. A2. We observe that GPT-4 evaluation is consistent with human experts in most cases, showcasing its reliability as a proxy of expert evaluators in saving evaluation costs. However, it occasionally misclassifies *Accurate* predictions into *Relevant*, and *Relevant* or *Opposite* predictions into *Irrelevant*, which we attribute to the fact that GPT-4 tends to favor more fluent answers instead of more informative ones. We leave more realistic and labor-saving evaluation strategies for future exploration.

### C.4 Multi-round Optimization

We incorporate the following datasets from [86] for multi-round fitness optimization:

- **Adeno-associated Viruses (AAV)** [91]. The dataset involves a 28-amino acid segment of the *caspid protein VP1* from *Adeno-associated virus*. The optimization objective is to improve its capability as a gene delivery vector.

- **Aliphatic Amide Hydrolase (AMIE)** [92]. The dataset aims to improve the enzymic activity of *Aliphatic amidase* from *Pseudomonas aeruginosa* in catalyzing the hydrolysis of short-chain aliphatic amides.

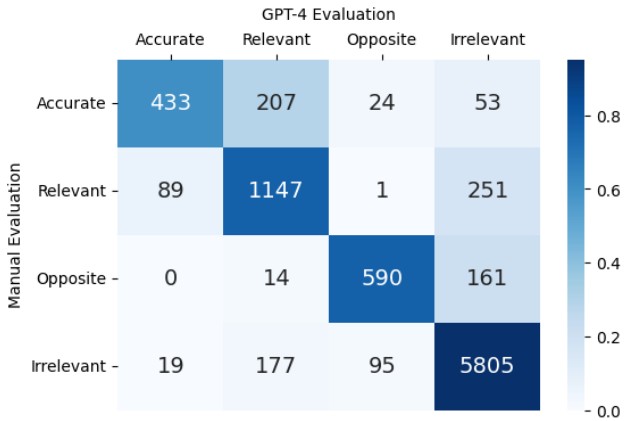

Figure A2: **Confusion matrix between GPT-4 and manual evaluation.**

Table A9: **Prompts for navigating mutation engineering.**

| Dataset | Prompt |
|---------|--------|
| AAV | Increased viability for packaging of a DNA payload for gene therapy. |
| AMIE | Increase in activity. |
| avGFP | Leads to enhanced fluorescence at 37 degrees Celsius. |
| E4B | Enhances cleavage by caspase-6 and granzyme B. |
| LGK | Increased enzyme activity. |
| UBE2I | Increased growth rescue rate at high temperature in a yeast strain. |

- **Green Fluorescent Proteins (avGFP)** [93]. The dataset aims to enhance the fluorescent intensity of the *Green Fluorescent Protein* from *Aequorea victoria*. The protein is widely adopted as a biosensor for detecting gene expressions and protein locations.

- **Ubiquitination Factor Ube4b (E4B)** [94]. The dataset aims to improve the enzymic activity of *Ubiquitin conjugation factor E4B* in *Homo sapiens*, which plays a role in proteasomal degradation by interacting with other proteins.

- **Levoglucosan Kinase (LGK)** [95]. The dataset focuses on *Levoglucosan kinase* in *Lipomyces starkeyi*. The optimization objective is to enhance its catalytic activity in canonical kinase phosphotransfer reaction.

- **SUMO E2 conjugase (UBE2I) [96].** The dataset studies *SUMO-conjugating enzyme UBC9* in *Homo sapiens* which is relevant to several human diseases. The optimization objective is to improve the growth rescue rate at high temperatures in a yeast strain.

We manually write prompts in Tab. A9 to navigate the optimization process by a beam search process. Specifically, we initialize the candidate set with the wild-type sequence. Then, for each round of optimization, we feed each candidate sequence and the textual instruction into the decoding workflow of MutaPLM. Then we sample $K$ mutations, the probability of which is proportional to the logits of the position head and the logits of the LM head. The optimization process is further detailed in Algorithm 1. The baselines are implemented by the EvoProtGrad [87] package. We perform experiments for 20 times, each comprising 10 optimization rounds.

## D    Additional Experiment Results

### D.1    Experiment Results on Temporal Split

The experimental results for mutation explanation and engineering are shown in Tab. A10 and Tab. A11 respectively. We observe that: (1) MutaPLM achieves promising performance on the temporal split and outperforms strong baselines, showcasing its robustness in handling novel mutations. (2) For mutation explanation, the experiment results are similar to those on the *Hard* set of the structural split, and we observe similar over-fitting issues as in structural split that more training steps lead

**Algorithm 1** Multi-round Optimization with Beam Search

---

**Require:** Wild-type Sequence $x_{\text{wt}}$, Instruction $t$, Number of Rounds $N$, Number of Candidates $K$
   $C \leftarrow \{x_{\text{wt}}\}$
   **for** Round $= 1, 2, \cdots, N$ **do**
      **for** $x \in C$ **do**
         $h \leftarrow f_{\text{PLM}}(x)$
         $h \leftarrow h + Decoder(h, T)$                    ▷ Add mutational features
         $\text{Score}^{\text{pos}}, \text{Score}^{\text{aa}} \leftarrow f_{\text{pos}}(h), f_{\text{LM}}(h)$      ▷ Calculate the logits two prediction heads
         $\text{Score}(x, i, j) \leftarrow \text{Score}^{\text{pos}}_i + \text{Score}^{\text{aa}}_{i,j}, \forall i \neq j$    ▷ The score mutating $i$-th amino acid to $j$
      **end for**
      $P(x, i, j) \leftarrow \text{GlobalSoftMax}[\text{Score}(x, i, j)]$    ▷ Probality distribution of sampling mutations
      $C \leftarrow \text{Mutate}(x, i, j), (x, i, j) \sim \text{SampleK}(P)$          ▷ Sampling without replacement
   **end for**
   **return** $C$

---

Table A10: **Performance evaluation for mutation explanation on temporal split.**

| Model | BLEU-2 | BLEU-4 | METEOR | ROUGE-1 | ROUGE-2 | ROUGE-L |
|---|---|---|---|---|---|---|
| ProLLaMA [45] | 0.69 | 0.21 | 3.33 | 0.83 | 0.04 | 0.80 |
| Galactica-6.7B [74] | 3.50 | 1.31 | 5.61 | 7.44 | 0.85 | 6.17 |
| Mol-Instructions [67] | 0.58 | 0.08 | 4.90 | 5.41 | 0.13 | 4.55 |
| GPT-4-0613 (5-shot, kNN) [33] | 9.30 | 4.25 | 15.08 | 13.92 | 2.29 | 11.84 |
| AugmentedESM [27] | 7.00 | 3.12 | 11.29 | 12.03 | 2.84 | 10.12 |
| Fine-tuned ESM-2 [19] | 6.90 | 3.83 | 13.86 | 14.21 | 4.63 | 12.62 |
| MutaPLM | **10.83** | **6.15** | **17.84** | **18.99** | **6.92** | **16.51** |

to improved validation loss but performance drops on the test set. This further underscores the significance of improving the generalization capability of mutation explanation models to assist real-world applications. (3) For mutation engineering, the results show little difference with those on the structural split. As discussed in Sec. 4.3, the PLM may have witnessed the protein sequence during pre-training, which mitigates the overfitting problem.

## D.2   Low-N Fitness Regression

While MutaPLM is not specifically designed for numeric tasks, we investigate if the learned *Delta* features could benefit fitness regression. We perform experiments on two protein fitness benchmarks, namely Spike-ACE2 [97] and avGFP [93]. Spike-ACE2 is a deep mutational scanning dataset that aims to predict the binding strengths between SARS-Cov-2 variants and its receptor ACE2, which is critical for identifying potentially dangerous strains of the virus. The avGFP benchmark aims to predict the fluorescence intensity of GFP variants, which is beneficial for developing biomarkers.

Table A11: **Performance evaluation for mutation engineering on temporal split.**

| Model | Accuracy (%) | Recall@50 (%) |
|---|---|---|
| Random | 4.40 | 0.81 |
| ProtST (ESM-2) [42] | 5.11 | - |
| GPT-4-0613 (5-shot, kNN) [33] | 12.13 | 6.28 |
| ESM-2 [19] | 34.76 | 24.02 |
| OntoProtein [75] | 37.74 | 28.49 |
| Fine-tuned BioMedGPT [62] | 34.57 | 4.09 |
| Fine-tuned ESM-2 [19, 83] | 55.78 | 44.04 |
| MutaPLM | **58.50** | **46.05** |

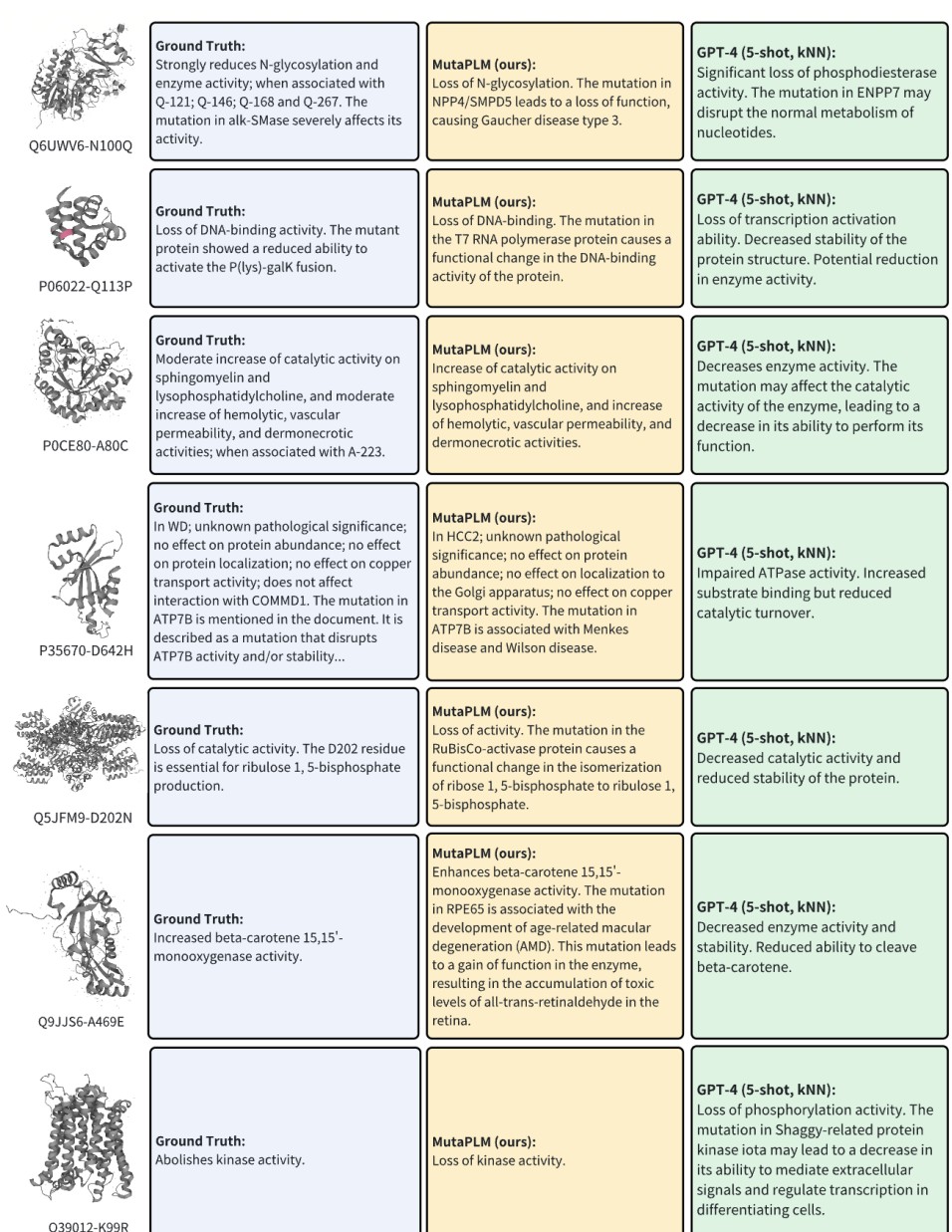

**Q6UWV6-N100Q**

**Ground Truth:**
Strongly reduces N-glycosylation and enzyme activity; when associated with Q-121; Q-146; Q-168 and Q-267. The mutation in alk-SMase severely affects its activity.

**MutaPLM (ours):**
Loss of N-glycosylation. The mutation in NPP4/SMPD5 leads to a loss of function, causing Gaucher disease type 3.

**GPT-4 (5-shot, kNN):**
Significant loss of phosphodiesterase activity. The mutation in ENPP7 may disrupt the normal metabolism of nucleotides.

**P06022-Q113P**

**Ground Truth:**
Loss of DNA-binding activity. The mutant protein showed a reduced ability to activate the P(lys)-galK fusion.

**MutaPLM (ours):**
Loss of DNA-binding. The mutation in the T7 RNA polymerase protein causes a functional change in the DNA-binding activity of the protein.

**GPT-4 (5-shot, kNN):**
Loss of transcription activation ability. Decreased stability of the protein structure. Potential reduction in enzyme activity.

**P0CE80-A80C**

**Ground Truth:**
Moderate increase of catalytic activity on sphingomyelin and lysophosphatidylcholine, and moderate increase of hemolytic, vascular permeability, and dermonecrotic activities; when associated with A-223.

**MutaPLM (ours):**
Increase of catalytic activity on sphingomyelin and lysophosphatidylcholine, and increase of hemolytic, vascular permeability, and dermonecrotic activities.

**GPT-4 (5-shot, kNN):**
Decreases enzyme activity. The mutation may affect the catalytic activity of the enzyme, leading to a decrease in its ability to perform its function.

**P35670-D642H**

**Ground Truth:**
In WD; unknown pathological significance; no effect on protein abundance; no effect on protein localization; no effect on copper transport activity; does not affect interaction with COMMD1. The mutation in ATP7B is mentioned in the document. It is described as a mutation that disrupts ATP7B activity and/or stability...

**MutaPLM (ours):**
In HCC2; unknown pathological significance; no effect on protein abundance; no effect on localization to the Golgi apparatus; no effect on copper transport activity. The mutation in ATP7B is associated with Menkes disease and Wilson disease.

**GPT-4 (5-shot, kNN):**
Impaired ATPase activity. Increased substrate binding but reduced catalytic turnover.

**Q5JFM9-D202N**

**Ground Truth:**
Loss of catalytic activity. The D202 residue is essential for ribulose 1, 5-bisphosphate production.

**MutaPLM (ours):**
Loss of activity. The mutation in the RuBisCo-activase protein causes a functional change in the isomerization of ribose 1, 5-bisphosphate to ribulose 1, 5-bisphosphate.

**GPT-4 (5-shot, kNN):**
Decreased catalytic activity and reduced stability of the protein.

**Q9JJS6-A469E**

**Ground Truth:**
Increased beta-carotene 15,15'-monooxygenase activity.

**MutaPLM (ours):**
Enhances beta-carotene 15,15'-monooxygenase activity. The mutation in RPE65 is associated with the development of age-related macular degeneration (AMD). This mutation leads to a gain of function in the enzyme, resulting in the accumulation of toxic levels of all-trans-retinaldehyde in the retina.

**GPT-4 (5-shot, kNN):**
Decreased enzyme activity and stability. Reduced ability to cleave beta-carotene.

**Q39012-K99R**

**Ground Truth:**
Abolishes kinase activity.

**MutaPLM (ours):**
Loss of kinase activity.

**GPT-4 (5-shot, kNN):**
Loss of phosphorylation activity. The mutation in Shaggy-related protein kinase iota may lead to a decrease in its ability to mediate extracellular signals and regulate transcription in differentiating cells.

Figure A3: **More case studies at mutation explanation.** We report the outputs of MutaPLM and GPT-4 (5-shot, kNN).

Table A12: **Performance evaluation on protein fitness regression benchmarks.** We perform experiments 5 times with different random seeds and report the Spearman correlation coefficient. The best and second-best results are marked in bold and underlined.

| Model | Spike-ACE2 | avGFP |
|---|---|---|
| Ridge Regression | $0.335\pm0.052$ | $0.298\pm0.071$ |
| ESM-2 [19] | $0.331\pm0.041$ | $0.554\pm0.013$ |
| Augmented ESM [27] | $0.363\pm0.021$ | $0.497\pm0.096$ |
| Augmented EVmutation [48] | $0.354\pm0.044$ | $0.512\pm0.034$ |
| ConFit [28] | $0.412\pm0.033$ | $0.564\pm0.035$ |
| Tranception_L [99] | $\mathbf{0.488\pm0.040}$ | $\underline{0.594\pm0.019}$ |
| MutaPLM | $\underline{0.481\pm0.028}$ | $\mathbf{0.596\pm0.032}$ |

Following prior works [98, 28], we adopt the low-$N$ setting with 192 randomly sampled training samples and 48 validation samples. We calculate the adaptive fitness by our PLM following [26] and concatenate it with the *delta* features $z_\Delta$. The result is fed into a trainable 2-layer MLP to predict the fitness scores, and the remaining parameters are kept frozen. We also implement baselines including Ridge Regression, ESM-2 [19], AugmentedESM [27], Augmented EVmutation [48], ConFit [28], and Tranception_L [99]. All the models are trained for 50 epochs with a batch size of 16 and a learning rate of 0.001 using the MSE loss. We sample different low-$N$ datasets with 5 random seeds and report the results in Tab. A12.

We observe that MutaPLM significantly outperforms baseline models that adopt ESM-2 as the PLM, indicating that the *delta* features have captured mutational knowledge from natural language supervision that benefits fitness regression tasks. While MutaPLM achieves comparable results with Tranception_L on both benchmarks, it is worth noting that the model adopts a different network architecture specifically designed for fitness regression. Therefore, we speculate that adopting a mutation-oriented PLM instead of ESM-2 may further improve the performance. While fitness regression is not the main focus of our work, we expect future endeavors that jointly harvest discrete textual descriptions and continuous fitness scores.

## D.3 Additional Case Studies

We present more case studies of mutation explanation in Fig. A3.

